# Yaws recurrence in children at continued risk of infection

**Camila G. Beiras**[1] *, **Michael Marks**[2,3], **Llorenç Quintó**[4,5], **Sergi Gavilán**[1], **Reman Kolmau**[6], **Maria Ubals**[1], **Marti Vall-Mayans**[1], **Oriol Mitjà**[1,6,7]

**1** STI and Skin NTDs Unit, Department of Infectious Diseases and Fight AIDS and Infectious Diseases Foundation, Hospital Universitari Germans Trias i Pujol, Badalona, Spain, **2** Clinical Research Department, London School of Hygiene and Tropical Medicine, London, United Kingdom, **3** Hospital for Tropical Diseases, University College London Hospitals, London, United Kingdom, **4** ISGlobal, Hospital Clínic-Universitat de Barcelona, Barcelona, Spain, **5** Manhiça Health Research Institute (CISM), Manhiça, Mozambique, **6** Lihir Medical Centre, International SOS, Londolovit, Papua New Guinea, **7** School of Medicine and Health Sciences, University of Papua New Guinea, Port Moresby, Papua New Guinea

* cgonzalez@flsida.org

**Data Availability Statement:** All relevant data are within the manuscript and its Supporting Information files.

**Funding:** The authors received no specific funding for this work.

## Abstract

### Background

In yaws-endemic areas, children with *Treponema pallidum subsp. pertenue* infection may suffer recurrent episodes due to either reinfection or relapse. However, the possibility of infection with other cutaneous ulcer causative agents and difficulties in interpreting standard laboratory results challenges the estimation of yaws recurrence rates.

### Methods

We estimated the rates of yaws recurrences in the Lihir Island (Papua New Guinea) using two approaches: passive surveillance based on a retrospective screening of electronic medical records of cutaneous ulcers diagnosed using serological testing between 2005 and 2016, and active surveillance conducted during a cross-sectional prevalence study which included PCR analyses of ulcers of all suspected cases of yaws. The risk of recurrent infection was assessed based on data from the passive surveillance analysis and using two Cox regression models (crude and multivariate), stratified by year of index episode. Data gathered from the active surveillance was used to characterize the recurrences and no hypothesis testing was performed.

### Results

The electronic medical records included 6,125 patients (7,889 ulcer episodes) with documented serological results of cutaneous ulcers of which 1,486 were diagnosed with yaws. Overall, 1,246/6,125 patients (20.3%) presented more than once with a cutaneous ulcer, and 103/1,486 (6.7%) patients had multiple episodes of yaws. The risk of yaws recurrence significantly increased with age and was higher in patients with ≥3 recurrent episodes. In the active surveillance, we identified 50 individuals with recurrent cutaneous ulcer that had PCR results available for both the index and recurrent episode. Of 12 individuals with *T.*

**Competing interests:** The authors have declared that no competing interests exist.

*pallidum* in the index ulcer, 8 (66%) had *T. pallidum* in subsequent assessments, relapse related to macrolide-resistance was identified in two of these cases.

## Conclusions

Our results confirm the need for active follow-up of yaws patients after treatment, particularly children and individuals with a history of recurrence.

## Author summary

Yaws is a neglected tropical disease produced by *Treponema pallidum* pertenue that causes skin ulcers in children living in remote rural areas of the South Pacific and West Africa. Children aged 5–15 and individuals with a history of recurrence are at higher risk of reinfection. Although yaws can be treated with single-dose azithromycin, some children present with recurrent cutaneous ulcers; however, the prevalence and risk factors for yaws recurrence are poorly known. Our analysis of skin ulcers in Papua New Guinea revealed that up to 20% of patients who presented to a health care facility in the Lihir island of Papua New Guinea with a cutaneous ulcer experienced a recurrent episode within the 6–36 months following treatment. Nearly all individuals with recurrent yaws were children aged 15 years or younger. Besides age, the number of previous ulcers was associated with a higher risk of recurrence. The molecular analysis revealed that among cases with T. pallidum at baseline that had a recurrence, this was often related to reinfection with the same microorganism. Our results confirm the need for active follow-up of young children diagnosed with cutaneous ulcers, with particular attention to those with younger age and previous history of recurrences.

## Introduction

Yaws, caused by *Treponema pallidum* subsp. *pertenue*, is an important public health problem in many countries worldwide. The disease predominantly affects the skin and bone, and most cases occur in children. Following treatment, skin lesions heal and titres of serological markers of infection, such as the Rapid Plasma Reagin (RPR), fall. However, in yaws-endemic areas, children present with recurrent cutaneous ulcers from either reinfection caused by ongoing contact with other infected individuals in their community or, more rarely, caused by relapse, often due to antibiotic-resistance [1]. Because yaws is commonly diagnosed using serology, reinfection is indistinguishable from relapse.

Epidemiological reports of yaws, in which cases of early active disease are rarely found in adults, suggest that human beings may develop some degree of immunity to reinfection [2]. Studies in experimental models have shown that animals experimentally infected with *T. pallidum* and subsequently treated can develop some degree of acquired immunity when re-challenged [3,4]. In this context, protection against reinfection is influenced by the duration of the original infection prior to treatment and seemed not to change after curative treatment with a lapse of time for re-exposure [5]. At the same time, within-host immune evasion by *T. pallidum* is well documented, and humans may experience multiple clinical episodes of syphilis and yaws, suggesting that immunity is non-sterilising and may be either transient or potentially strain-specific [6–8]. Better understanding the risk of recurrence of yaws may help shed light both on immunity to *T.pallidum* and inform programmatic surveillance efforts.

There is very little clinical data about whether natural reinfection with *T. p. pertenue* occurs after treatment. The clinical picture is complicated because of the occurrence of latent yaws infection, during which individuals have reactive serology but no clinical evidence of the disease. In yaws-endemic regions, children may present with skin ulcers caused by a range of etiological agents; therefore, serological tests cannot differentiate between active infectious ulcer caused by *T. p. pertenue* and a co-infection with latent yaws and a cutaneous ulcer caused by another agent. In particular, *Haemophilus ducreyi*, a gram-negative bacteria, has been found to be a major cause of skin ulcers in yaws endemic regions of the tropics [9] and is commonly found in individuals with reactive serology for yaws.

In this study, we used data from both passive and active surveillance to identify rates of recurrence, of cutaneous ulceration, either reinfection or relapse, due to yaws and other aetiologies in a highly endemic region of Papua New Guinea.

## Methods

### Ethics statement

The Medical Research Advisory Committee of the Papua New Guinea National Department of Health approved the study protocol (number 12.36). All participants (or their parent or guardian in case of children) provided oral informed consent. Additionally, written informed consent was obtained from individuals with suspected active yaws before the etiologic study. In case of children, the parent or guardian provided written consent and the children consented orally.

### Overview of study setting

The study was conducted in Lihir Island in the New Ireland Province of Papua New Guinea. According to 2016 census data, Lihir has an estimated population of 22,000 inhabitants. In Lihir, public healthcare is delivered through aid posts spread across the island attended by nurses and the Lihir Medical Centre (LMC), a modern private rural hospital run by physicians and nurses. Aid posts lack essential laboratory tests; therefore, diagnose skin diseases clinically only, whereas the LMC can perform tests, such as the RPR. Diagnoses and treatments of skin ulcers performed out of public health campaigns are afforded by patients. We previously estimated that the overall incidence of yaws in the population was 1% per year, with an incidence among children aged between 1–15 years of 7% [10]. The study objectives were to determine the overall incidence of recurrences and to characterize the aetiology patterns of recurrences. Therefore, we used two approaches for case detection: passive surveillance based on screening of hospital electronic records and active surveillance of skin ulcers during a public health intervention for yaws elimination that included molecular analyses of skin ulcers.

### Passive surveillance

We retrospectively screened hospital electronic medical records from the LMC to identify cases of recurrent yaws. Although yaws can be treated clinically in other aid posts spread across the island, serological analyses can only be performed at LMC. All outpatients diagnosed with infectious cutaneous ulcers of any aetiology between Jan 1, 2005 and Jun 1, 2016 were screened.

A case of cutaneous ulcer was identified by the presence of the diagnostic code listed in either the first or second diagnostic position of an outpatient medical encounter (ICD-9 codes 707.1 and 707.9). We excluded cases of non-infectious ulcers, including diabetic, venous, and post-trauma ulcers. Serological testing is the routine diagnostic method to confirm yaws at

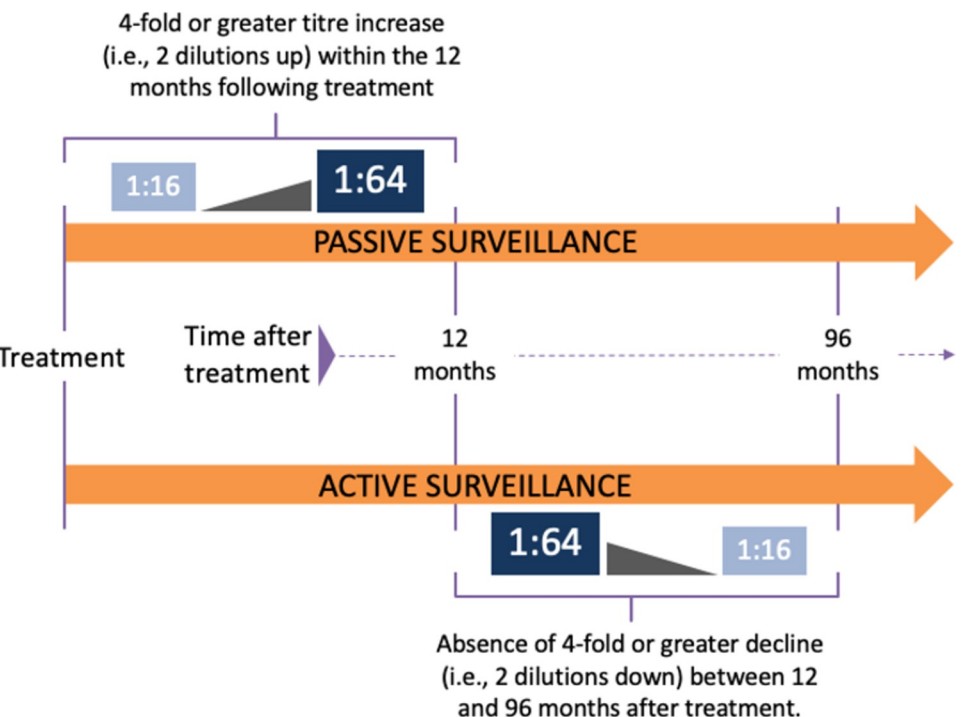

**Fig 1. Definition of recurrence in the passive and active surveillance analysis.**

LMC. In this centre, a presentation of cutaneous ulcer together with a positive *Treponema pallidum* haemagglutination assay (TPHA) and RPR ≥ 1:8 are considered diagnostic of active yaws. Cutaneous ulcers with recorded RPR test results < 1:8 are classified as non-yaws ulcers. Ulcer cure is not routinely confirmed by RPR test, and patients are not actively followed-up after treatment. Therefore, we considered recurrent yaws in all individuals who had a previous entry in the electronic records with serologically confirmed yaws ulcer, achieved clinical remission after treatment, and experienced a recurrent episode of cutaneous ulcer that met one of the following criteria of RPR titre change compared with the index episode: (1) 4-fold or greater titre increase within the 12 months following treatment or (2) absence of 4-fold or greater decline between 12 and 96 months after treatment (Fig 1).

We retrieved clinical information about every patient on a specifically designed database, including the date of visit, RPR titre, beginning of follow-up, months since previous ulcer episode, number and date of ulcer episodes, months since previous yaws episode, and number and date of yaws episodes. We also retrieved demographic information, including age, gender, the village of origin, and distance to the hospital. We assessed the accessibility to the hospital based on the distance between the patient's place of residence and the hospital. Roads in Lihir are non-paved tracks, and the land is covered with woody vegetation. The vehicle travelling time from the remotest village is 1.5 hours with longer times during the rainy season. The distance (Km) between a given village and healthcare centre was grouped as follows: 0 to <5, 5 to <15, 15 to <25, and ≥25 Km.

## Active surveillance

We used data collected as part of a prospective study of yaws elimination [1]. Briefly, the study consisted of repeated clinical surveys for active yaws, serologic surveys for latent yaws, and

molecular analyses; surveys were conducted before and 6 and 12 months after mass drug administration (MDA) with azithromycin. During the initial MDA and at each subsequent survey, we swabbed the ulcers of all suspected cases of yaws for polymerase chain reaction (PCR) testing. Antimicrobial therapy with single dose azithromycin 30mg/Kg up to 2g was administered at the time of each visit. We defined molecularly confirmed recurrence when an individual had a new episode of PCR-confirmed yaws after clinical remission in the previous survey (Fig 1). The following possible scenarios of repeated episodes of cutaneous ulcers where also considered: *T. pertenue* DNA detected in the index episode but not in the second one, *T. pertenue* DNA detected in the second episode but not in the first one, and no *T. pertenue* DNA detected in either of the two episodes.

## Statistical analysis

Variables collected from outpatient records were described as medians, interquartile ranges, and percentages. Patients were considered to be at risk for recurrence from the day of an index episode until the end of the study period (01 Jun 2016). Serologically-confirmed recurrent episodes were identified by passive case detection, assuming no migrations or deaths during the follow-up period. Crude and multivariable Cox conditional regression models applying the Prentice, Williams, and Peterson approach (PWP) were estimated to assess the risk for recurrent infections per subject over time [11]. The models, built using a robust variance estimate, were stratified by the year of index episode, allowing the baseline hazard function to differ throughout the entire follow-up period, which is appropriate for long periods that may be affected by environmental, political, economic changes as well as by the health-seeking behaviour of the population. Although the models estimated with the PWP method are recommended to stratify according to the number of previous episodes, we used this information as a covariate in the model to account for change in the hazard for each additional episode. Since the number of previous episodes (ulcers or yaws) and the time between episodes changed over time during the follow-up period, we tested for time-varying coefficients of those covariates in the adjusted model. Interactions between the number of episodes and elapsed time between episodes were also considered. Analyses regarding the findings of active surveillance were descriptive and no hypothesis testing was performed. We used the same recurrence criteria as for the case definition in passive surveillance analysis (i.e., 4-fold or greater RPR titre increase). All analyses, data manipulation, and implementations were done in Stata ver16 [12].

## Results

### Passive case detection

Overall, 8,598 diagnoses of cutaneous ulcers were reported in the electronic records from LMC during the study period (Fig 2A). We excluded 388 episodes that lacked documented serological results, 139 patients with non-infectious ulcers, and 182 that lacked a documented date of birth. The resulting study dataset consisted of 7,889 episodes among 6,125 patients. Of them, 1,486 patients were classified as having had a serologically-confirmed yaws ulcer. Overall, 1,246/6,125 (20.3%) individuals presented more than once to LMC with a cutaneous ulcer, and 103/1,486 (6.9%) presented more than once with serologically-confirmed yaws. The demographic and clinical characteristics of patients at diagnosis of the primary and recurrent yaws episodes were similar (Table 1). Patients had a median age of 9.7 years, 57% were male, and more than half had an RPR titre between 1:32–1:64.

According to the multivariate analysis, the occurrence of three or more previous ulcer episodes was significantly associated with the risk of recurrence (Table 2). The hazard ratio (HR) of yaws recurrence decreased progressively with age, being children in the 0 to <5 years group

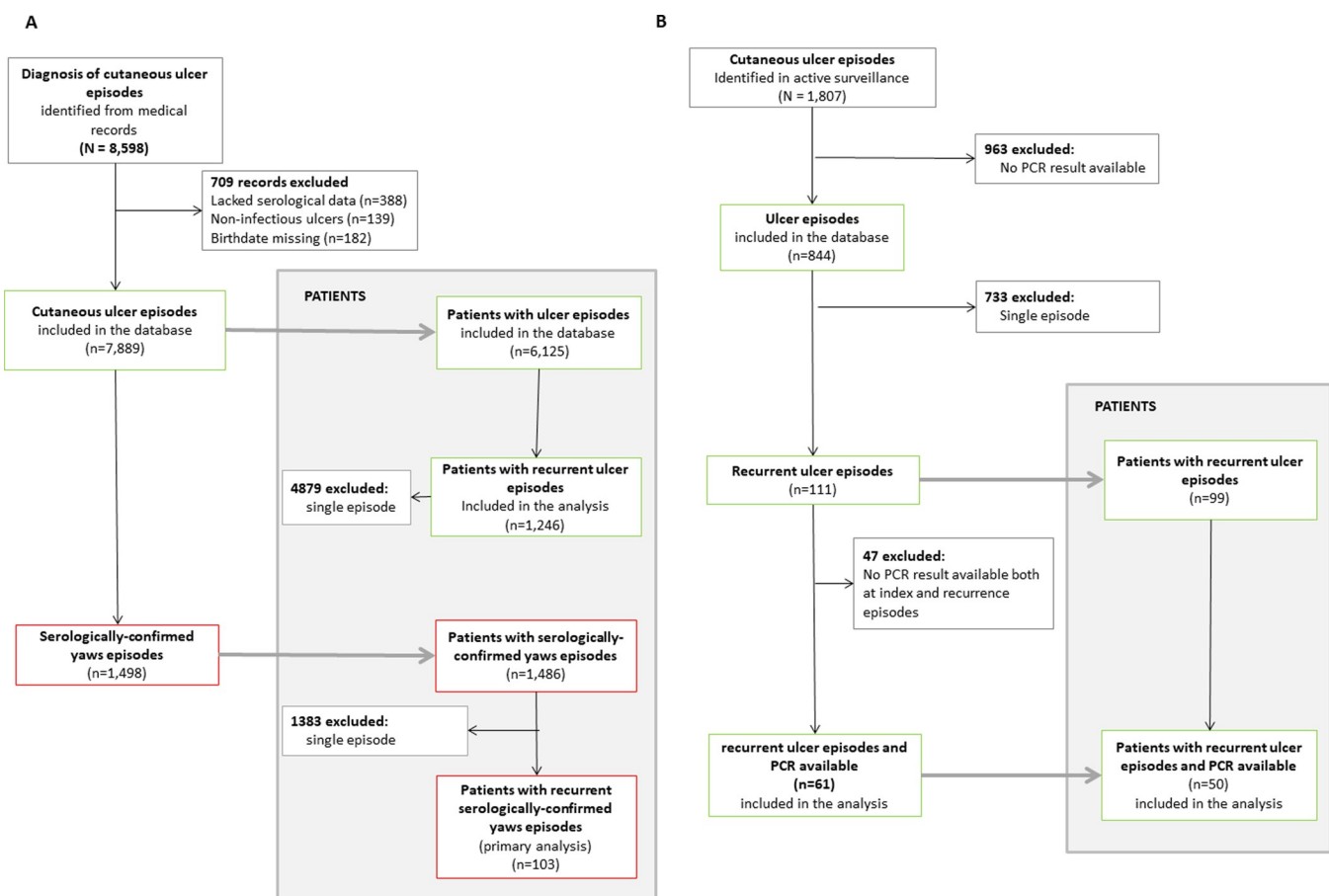

**Fig 2. Study profile A: Passive case detection through hospital records review; cases were confirmed by a serological test.** B: Active case detection in mass drug administration setting; cases were confirmed by PCR.

associated with the highest risk of recurrence. The time-varying coefficient analysis showed that the effect of the number of previous episodes and time since the last episode did not change over time (HR 1.00, $p = 0.81$) for all the covariates. Likewise, we did not find meaningful interactions between time from the last episode and the number of previous episodes, neither between the time since the last yaws episode, and time from the last ulcer. Distance to the hospital and the RPR titre at primary episode did not show a significant contribution to the multivariate model.

## Active case detection

During nine biannual community surveys, we diagnosed 1,701 individuals with 1,807 cutaneous ulcers; lesion samples from 844 (46.7%) cutaneous ulcers were tested by PCR. Ninety-nine (5.8%) individuals had a total of 111 recurrent episodes: 90 individuals with one recurrent ulcer, six individuals with two recurrences, and three individuals with three recurrences. (Fig 2B) PCR results of lesion swabs for both the index and recurrent episodes were available for 50/99 individuals, who accounted for 61/111 recurrent ulcers. Table 3 shows the aetiology of the index and recurrent episodes of cutaneous ulcer. Of 12 individuals with *T. pallidum* in the index ulcer (alone or dual infection), 8 (66.6%) had *T. pallidum* in subsequent assessments. Correspondingly, of 33 people with *H. ducreyi* in the index ulcer (alone or dual infection), 20 (60.6%) had *H. ducreyi* in subsequent assessment.

**Table 1. Characteristics of patients included in the passive case detection analysis (i.e., retrospective screening of electronic health records).**

| | | Number of Yaws-episode | | | |
|---|---|---|---|---|---|
| | | Primary | First recurrence | Second recurrence | Third recurrence |
| | | (N = 1486) | (N = 103) | (N = 7) | (N = 2) |
| Male [a] | | 850 (57%) | 56 (54%) | 6 (86%) | 1 (50%) |
| Age (years) [b] | | 9.7 (6.1–14.0) | 10.2 (7.8–11.9) | 9.9 (9.2–12.8) | 11.4 (10.1–12.6) |
| Age (years) [a] | 0-<5 | 248 (17%) | 9 (9%) | 0 (0%) | 0 (0%) |
| | 5-<15 | 918 (62%) | 85 (83%) | 7 (100%) | 2 (100%) |
| | 15-<25 | 167 (11%) | 6 (6%) | 0 (0%) | 0 (0%) |
| | ≥25 | 153 (10%) | 3 (3%) | 0 (0%) | 0 (0%) |
| Distance to hospital [a] | <5Km | 588 (40%) | 50 (49%) | 5 (71%) | 2 (100%) |
| | 5-10Km | 299 (20%) | 26 (25%) | 1 (14%) | 0 (0%) |
| | >10Km | 283 (19%) | 16 (16%) | 1 (14%) | 0 (0%) |
| | Unknown | 316 (21%) | 11 (11%) | 0 (0%) | 0 (0%) |
| RPR titre at primary episode [a] | 1:8 to 1:16 | 558 (38%) | 22 (21%) | 0 (0%) | 0 (0%) |
| | 1:32 to 1:64 | 771 (52%) | 66 (64%) | 6 (86%) | 2 (100%) |
| | 1:128 or higher | 157 (11%) | 15 (15%) | 1 (14%) | 0 (0%) |
| Number of previous ulcer episodes [a] | 0 | 1348 (91%) | 0 (0%) | 0 (0%) | 0 (0%) |
| | 1 | 114 (8%) | 76 (74%) | 0 (0%) | 0 (0%) |
| | 2 | 18 (1%) | 22 (21%) | 2 (29%) | 0 (0%) |
| | ≥3 | 6 (0%) | 5 (5%) | 5 (71%) | 2 (100%) |
| Months since index yaws episode [b] | | | 25.3 (18.9–41.0) | 50.0 (37.6–52.5) | 61.5 (49.2–73.7) |
| Months since previous ulcer episode [c] | | 18.2 (6.6–37.2) [138] | 23.7 (13.9–33.8) [103] | 24.8 (15.3–27.5) [7] | 16.4 (11.6–21.2) [2] |
| Months since previous yaws episode [b] | | | 25.3 (18.9–41.0) | 26.6 (15.3–31.6) | 16.4 (11.6–21.2) |

[a] n (Column percentage) of the various categories of variables distribution

[b] Median (IQR)

[c] Median (IQR) [n]

Of eight patients with PCR-confirmed recurrent yaws, 3 (37.5%) had evidence of serological failure (i.e., ≥4-fold increase in RPR titre), 4 (50.0%) increased RPR titre, albeit not reaching the recurrence criteria, and 1 (12.5%) had a 4-fold reduction in RPR titre consistent with serological cure (Table 4). All eight patients with recurrent yaws had wild type 23S rRNA sequences in the index ulcer. According to the PCR analysis of recurrent ulcers, two cases had A2059G mutations associated with resistance to macrolide, and five were wild-type strains; the 23s rRNA gene could not be amplified in one case.

## Discussion

In this study, we found that up to 20% of patients who presented to a health care facility in the Lihir island of Papua New Guinea with a cutaneous ulcer experienced recurrent episode within the 6–36 months following treatment. A low–albeit substantial (7%)–proportion of patients with yaws had serologically-confirmed recurrent yaws. Nearly all individuals with recurrent yaws were children aged 15 years or younger. To date, the epidemiology of ulcer recurrence has been barely investigated in yaws; however, our findings are in line with those reported for other treponemal infections such as syphilis (recurrence range from 4%- 9%) [13–16]. Our active search study, in which aetiology could be confirmed by PCR, revealed that among cases with *T. pallidum* at baseline that had a recurrence, this was often related to reinfection with the same microorganism.

**Table 2. Multivariate model for predicting recurrence among patients included in the passive case detection analysis of individuals from electronic health records.**

| Covariate | | Crude Model [a] | | | Adjusted Model [b] | | |
|---|---|---|---|---|---|---|---|
| | | Hazard Ratio (95%CI) | *p*-value | Overall p-value | Hazard Ratio (95%CI) | p-value | Overall p-value |
| Male | | 0.95 (0.64–1.41) | 0.7987 | | 0.87 (0.60–1.25) | 0.4363 | |
| Age (years) | 0-<5 | 1 | | 0.0011 | 1 | | 0.0009 |
| | 5-<15 | 0.69 (0.45–1.06) | 0.0928 | | 0.65 (0.44–0.97) | 0.0338 | |
| | 15-<25 | 0.26 (0.10–0.66) | 0.0045 | | 0.26 (0.10–0.65) | 0.0043 | |
| | ≥25 | 0.11 (0.03–0.45) | 0.002 | | 0.11 (0.03–0.45) | 0.0023 | |
| Distance hospital | <5Km | 1 | | 0.0273 | 1 | | 0.1439 |
| | 5-10Km | 0.82 (0.52–1.30) | 0.3955 | | 0.85 (0.55–1.31) | 0.4589 | |
| | >10Km | 0.53 (0.30–0.94) | 0.0292 | | 0.62 (0.37–1.05) | 0.0783 | |
| | Unknown | 0.43 (0.22–0.83) | 0.0118 | | 0.55 (0.29–1.04) | 0.0676 | |
| RPR titre at primary episode | 1:08 to 1:16 | 1 | | 0.2977 | 1 | | 0.1038 |
| | 1:32 to 1:64 | 0.74 (0.47–1.16) | 0.1867 | | 0.67 (0.43–1.02) | 0.0644 | |
| | 1:128 to 1:512 | 0.57 (0.23–1.42) | 0.2259 | | 0.50 (0.21–1.17) | 0.1085 | |
| Number of previous ulcer episodes | 0 | 1 | | <0.0001 | 1 | | 0.0208 |
| | 1 | 1.91 (1.20–3.04) | 0.0065 | | 1.46 (0.54–3.92) | 0.4524 | |
| | 2 | 2.56 (1.10–5.94) | 0.0289 | | 1.93 (0.66–5.64) | 0.2286 | |
| | ≥3 | 6.03 (2.78–13.09) | | | 4.13 (1.64–10.43) | 0.0026 | |
| Number of previous yaws episodes | 0 | 1 | | 0.0019 | 1 | | 0.714 |
| | 1 | 2.13 (1.29–3.51) | 0.003 | | 0.75 (0.23–2.48) | 0.6345 | |
| | ≥2 | 5.89 (1.88–18.46) | 0.0023 | | 1.14 (0.29–4.44) | 0.847 | |
| Months since previous ulcer episode [c] | | 1.02 (1.01–1.03) | 0.0001 | | 1.01 (0.97–1.04) | 0.7109 | |
| Months since previous yaws episode [c] | | 1.03 (1.02–1.04) | | | 1.01 (0.97–1.06) | 0.5047 | |

**RPR:** Rapid Plasma Reagin

[a] Cox regression stratified by year of primary episode (one model per covariate)

[b] Cox regression stratified by year of primary episode and forcing all covariates into the model

[c] Hazard Ratio per unit increase

Several observations in our study argue against the development of fully protective immunity in yaws. First, patients with a higher cumulative number of treated serologically-confirmed yaws ulcers had an increased hazard of subsequent serologically-confirmed yaws ulcers. This effect may be favoured by behavioural determinants, such as treated patients returning to environments with a high force of reinfection–therefore, increased likelihood of reinfection–, or individual-level risk factors (e.g., genetics) that predispose them to yaws disease. Second, we provide evidence that children are re-infected with the same strain of *T.p.*

**Table 3. Etiology of recurrent cutaneous ulcers according to the type of the index episode in patients identified during a public health intervention for yaws eradication (active case detection).** Data correspond to 61 ulcers detected among 50 individuals with PCR available.

| Index episode | Recurrent episode | | | | Total |
|---|---|---|---|---|---|
| | *T.p.pertenue* only detected | *H.ducreyi* only detected | Dual infection detected | Negative in all tests | |
| *T.p.pertenue* only detected | 4 | 0 | 0 | 0 | 4 |
| *H.ducreyi* only detected | 2 | 14 | 3 | 6 | 25 |
| Dual infection detected | 3 | 2 | 1 | 2 | 8 |
| Negative in all tests | 1 | 9 | 3 | 11 | 24 |
| Total | 10 | 25 | 7 | 19 | 61 |

**Table 4. Clinical and laboratory findings of 8 cases of PCR-confirmed yaws reinfection among individuals identified during a public health intervention for yaws eradication (active case detection analysis).**

| Study no | Age (years) | Gender | Type of lesion (size) | Duration of current episode in weeks | RPR titre | Strain genotype | Macrolide resistance | RPR nadir | Months between baseline and repeat yaws | Type of lesion (size) | Duration of current episode in weeks | RPR repeat | Strain genotype | Macrolide resistance |
|---|---|---|---|---|---|---|---|---|---|---|---|---|---|---|
| | | | Baseline episode of yaws | | | | | RPR nadir | Repeated episode of yaws | | | | | |
| 1 | 9 | F | ulcer (2.5 cm) | 3 | 1:08 | JG8 | WT | NR | 6 | papilloma (1 cm) | 12 | 1:16 | JG8 | WT |
| 4 | 2 | M | ulcer (3 cm) | 3 | 1:08 | JG8 | WT | No data | 6 | ulcer (5 cm) | 2 | 1:128 | JG8 | not amplified |
| 5 | 10 | F | ulcer (2 cm) | 3 | 1:64 | Not amplified | WT | No data | 6 | ulcer (2 cm) | 3 | 1:04 | JG8 | WT |
| 6 | 10 | M | ulcer (3 cm) | 3 | Negative | JG8 | WT | No data | 6 | ulcer (1 cm) | 1 | 1:32 | JG8 | WT |
| 7 | 10 | F | ulcer (1 cm) | 1 | 1:16 | JG8 | WT | No data | 6 | ulcer (2 cm) | 1 | 1:32 | JG8 | WT |
| 8 | 10 | M | ulcer (4 cm) | | 1:02 | JG8 | WT | NR | 12 | papilloma (1cm) | 6 | 1:64 | JG8 | WT |
| 2 | 10 | M | ulcer (2 cm) | 2 | 1:16 | JG8 | WT | No data | 6 | papilloma (3 cm) | 32 | 1:16 | JG8 | A2059G |
| 3 | 12 | M | ulcer (1 cm) | 2 | 1:08 | Not amplified | WT | NR | 30 | ulcer (3 cm) | 15 | 1:16 | JG8 | A2059G |

**M:** male. **F:** female. **NR:** not reported. **RPR:** Rapid Plasma Reagin. **WT:** wild-type

*pertenue* (e.g., JG8), arguing against strain-specific immunity. Of note, we have previously used whole-genome sequencing to demonstrate that strains based on multi-locus sequence typing (MLST) may be further sub-divided [17]. Therefore, we cannot exclude the possibility of sub-lineage-specific immunity. Lower rates of yaws in adults might be influenced by innate differences manifested with age, age-related behavioural exposure, and age-dependent immunological effects, which we could not detect in this study. Further data, such as establishing whether re-infection occurs with strains which are identical by whole genome sequencing, may be needed to assess whether acquired resistance to reinfection with yaws can develop independently of age-related behavioural factors. Our data also demonstrated repeated infection with *H. ducreyi* in consecutive episodes and suggests that patients do not develop immunity to *H. ducreyi*.

Our study has several limitations. Our results might underestimate the actual burden of recurrence because we could not ensure completeness of follow-up, and some subjects may have migrated, been attended in one of the peripheral aid-posts in the Island after the index episode or be reluctant (or have difficulties) to visit LMC. Second, a false-positive RPR test can occur and may have biased the estimate of recurrent yaws ulcers. We used a treponemal test together with the RPR test with a high titre threshold (>1:8) to define yaws, which is unlikely to be related to common causes of false-positive results [18]. Also, we cannot completely rule out treatment failures, which can occur because of multiple factors; however, considering the efficacy of azithromycin [19] and that therapy was administered under supervision, we did not expect treatment failures to account for a substantial contribution to new ulcers. Our passive case detection analysis is also based only on serologically-confirmed yaws which is not completely reliable. However, our findings are supported by our active case finding exercise where molecular confirmation was used.

Our results confirm the need for active follow-up of young children diagnosed with cutaneous ulcers, with particular attention to those with younger age and previous history of recurrences. This follow-up, currently conducted to monitor azithromycin resistances, should also identify recurrences, since it would provide an opportunity to identify contacts of yaws patients and plan interventions beyond the case. As a significant burden of yaws transmission may not all occur within the household [20,21], there is a need to develop effective strategies to ensure all household and community contacts are identified and treated to reduce the risk of reinfection and break chains of transmission within communities.

## Supporting information

**S1 Data. Dataset of active ulcer cases included in the analysis.**
(ODS)

## Acknowledgments

The authors would like to thank Gerard Carot-Sans for reviewing and editing the final draft of the manuscript and Eric Q. Mooring for his advice on data analysis.

## Author Contributions

**Conceptualization:** Camila G. Beiras, Michael Marks, Llorenç Quintó, Maria Ubals, Marti Vall-Mayans, Oriol Mitjà.

**Formal analysis:** Llorenç Quintó.

**Investigation:** Camila G. Beiras, Sergi Gavilán, Oriol Mitjà.

**Methodology:** Reman Kolmau, Maria Ubals.

**Project administration:** Camila G. Beiras, Sergi Gavilán, Reman Kolmau, Maria Ubals.

**Supervision:** Michael Marks, Marti Vall-Mayans, Oriol Mitjà.

**Writing – original draft:** Camila G. Beiras, Michael Marks, Maria Ubals, Oriol Mitjà.

**Writing – review & editing:** Camila G. Beiras, Michael Marks, Llorenç Quintó, Sergi Gavilán, Reman Kolmau, Maria Ubals, Marti Vall-Mayans, Oriol Mitjà.

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
