## [Decision Letter · Decision Letter 0]

20 Oct 2021

Dear Dr. González-Beiras,

Thank you very much for submitting your manuscript "Yaws recurrence in children at continued risk of infection" for consideration at PLOS Neglected Tropical Diseases and thank you for your patience during the longer than usual period for peer review. As with all papers reviewed by the journal, your manuscript was reviewed by members of the editorial board and by several independent reviewers. In light of the reviews (below this email), all reviewers found merit in your manuscript, and thus we would like to invite the resubmission of a significantly-revised version that takes into account the reviewers' comments. Particularly I recommend you pay close attention to the clarifications around ethics highlighted by Reviewer 3. 

We cannot make any decision about publication until we have seen the revised manuscript and your response to the reviewers' comments. Your revised manuscript is also likely to be sent to reviewers for further evaluation.

Sincerely,

Stuart Robert Ainsworth

Associate Editor

Ana LTO Nascimento

Deputy Editor

Reviewer's Responses to Questions

**Key Review Criteria Required for Acceptance?**

**Methods**

-Are the objectives of the study clearly articulated with a clear testable hypothesis stated?

-Is the study design appropriate to address the stated objectives?

-Is the population clearly described and appropriate for the hypothesis being tested?

-Is the sample size sufficient to ensure adequate power to address the hypothesis being tested?

-Were correct statistical analysis used to support conclusions?

-Are there concerns about ethical or regulatory requirements being met?

Reviewer #1: The objectives of the study were clearly tested. Hypothesis testing was not indicative for this study.

The study design was appropriate with good description of the study participants (some clarifications have been asked; included in the attachment) 

Sample size was adequate. Few suggestions for statistical analyses have been included (see attachment)

No major ethical issues. The authors should however provide a statement on ethical considerations for the study

Minor Revisions required

Reviewer #2: (No Response)

Reviewer #3: - The hypothesis is not clear stated in the background. The background misses stating the link about the knowledge gap on immunity to re-infection and how this study (estimating recurrence with these particularly methods) can contribute to it.

Explaining in the background, that these first estimations of recurrence (as a first step) can start pointing towards the type of immunity (saying that you will use of molecular methods), while giving clear recommendations on the type of surveillance and follow up needed (in the discussion-recommendations), may clarify the justification of conducting this study and its added value.

- The appropriateness of the study design could be better clarified, by explaining why these two methods are chosen (i.e. in the absence of a formal surveillance system), and how they correlate to each other (in time, in target study population), and the added value of putting both approaches together in a study.

- The study population is not clearly mentioned (and should be specified for the two surveillance approaches included). The study setting mentions Lihir island, but in the Passive surveillance description, it is only mentioned the Lihir Medical Centre (LMC) consulting patients, so it should be described which is the target population of LMC in comparison with Lihir population (i.e. also if there are any other health posts-centers in the island which were treating ulcers during the study period? and if different characteristics in access), if the treatment was for free for everybody (or certain age groups), etc. All this orients towards accessibility, and potentially completeness-representativeness of the passive surveillance data and its interpretations.

**Results**

-Does the analysis presented match the analysis plan?

-Are the results clearly and completely presented?

-Are the figures (Tables, Images) of sufficient quality for clarity?

Reviewer #1: Analysis plan and results clearly presented. 

Quality of the figure (flow diagram) provided should be improved 

Minor revisions requested (See detailed comments in attachment)

Reviewer #2: (No Response)

Reviewer #3: - Clarity of the results:

In the passive surveillance, are there any differences among patients with positive and negative serology (in terms of demography, number of ulcers, distance to centre)? 

- Clarity of figures and tables:

Title of figures and table should be self-explanatory (at this point they miss to inform about which how data was collected (if from passive-active or both), from where, when, etc)

To specify in the title of Figure 1 that relates to both active and passive surveillance (to clarify since it is inserted in the Passive surveillance section)

In Figure 1, the A and B in the graph (A is active surveillance) does not correspond with the legend (A is passive)

In Figure 1, in the Diagnoses of cutaneous ulcers from medical records, there is a group of patients who have PCR (not clear why), since in the section of the Methods only serological methods are defined in the Passive surveillance.

**Conclusions**

-Are the conclusions supported by the data presented?

-Are the limitations of analysis clearly described?

-Do the authors discuss how these data can be helpful to advance our understanding of the topic under study?

-Is public health relevance addressed?

Reviewer #1: The study has significant public health indications and offers good evidence on the need for follow-up of yaws cases. The conclusions are valid.

Some suggestions have been made with respect to study limitations (see attachment)

Minor revisions required

Reviewer #2: (No Response)

Reviewer #3: - The main conclusion stated is supported by the findings from both surveillance systems.

- Further details on the limitations of using such medical records as surveillance system should be mentioned (i.e. only completeness already mentioned, not any other of the surveillance attributes which may affect the number of recurrent consultations). 

- The authors suggest absence of full protective immunity, but they could explain which methodologies are recommended to properly confirm this argument.

- The public health relevance of the recommendation (conducting active follow up) is there, but it could be better integrated in the public health context (not as an isolated recommendation). For instance, it could be mentioned under which strategy (in the context of eradication plan) is recommended the active follow up for this/similar settings (i.e. active follow up embedded on a Lihi passive surveillance with ulcers as weekly notifiable disease? Active follow up under weekly or monthly active surveillance? Active follow up after regular MDAs?).

**Editorial and Data Presentation Modifications?**

Reviewer #1: Minor revision required (see reviewer's attachment)

Reviewer #2: (No Response)

Reviewer #3: Minor modifications:

- To clarify which diagnostic codes were included under the passive surveillance – so it is possible to ensure the quality of the case definition used for the objective of the surveillance

- Accessibility is reflected in the analysis as Km to the health center in the passive surveillance analysis. But there are other aspects related to accessibility which are not considered. For instance, it would be useful to know whether treatment was offered for free, and any information on the type of care provided (24 hours service, etc), acceptability by population of the healthcare provided.

- For the active surveillance data– It would be useful to describe (or refer to in the text – if published elsewhere) how the active finding was done (so reader can understand how systematic was the population targeting and the coverage) 

- It is not clear how a new episode of PCR-confirmed was defined. Was clinical remission after treatment followed, or any other time variables?

- The use of same recurrence criteria for the case definition in the active and passive surveillances should be highlighted in the methods (and not only at the results section, at the end of the active surveillance system). For clarity, it could be considered to have a definitions sub-section in the methods section.

**Summary and General Comments**

Reviewer #1: Overall, well written paper which adds significant evidence to the body of knowledge in the epidemiology of yaws and useful insights into recurrence of yaws.

Reviewer #2: Comments:

1. On the end of page 1/beginning of page w, the sentence beginning “studies in experimental models” is a bit confusing; can you clarify what “acquired protection to T. pallidum can develop in untreated animals and persist after treatment” means? What is meant by “acquired protection” – were they protected after a challenge? I think the next sentence explains it but it may be more clear just to drop that first sentence and expand on the theme of the paragraph – is it that there is a conflict between experimental models and humans in the generation of protective immunity? 

2. At the end of that same paragraph the sentence “humans may experience multiple episodes of syphilis and yaws, suggesting that immunity is either transient or potentially strain-specific” seems fairly limited in the interpretation of multiple infections. Even long-lived immune responses many not offer sterilizing immunity (see recent serological data on trachoma where kids get multiply infected yet have good antibody responses). 

3. Methods: Is it standard to have a definition of RPR titer >=1:8 as serological confirmation of yaws? Marks et al Lancet Global Health PMCID: PMC7116878 used an RPR titer of 1:2 as evidence of seroconversion, so it would be useful to give some justification for this decision. The discussion briefly calls this a high-titer cutoff (although the Marks et al paper seems to use 1:16 as a high titer cutoff – it would be helpful to have consensus on what RPR titer constitutes high titer).

4. Results, paragraph 1: 103/1486 does not = 60.7%. Assuming there is a typo somewhere here?

5. I am confused by the flow of Figure 1 – there seems to be an increasing number of people in the flow. IFor example, why is the flow going from a smaller number of patient with PCR available (50) to a larger number of patients with recurrent episodes (1246) with >4000 people excluded? 

6. Discussion. I am a little confused by the actual conclusions about the recurrence of yaws. Due to the error I noted above it is not clear what % of cases seen were recurrent yaws in the passive case detection section. In the active case finding section, 8/12 PCR-confirmed in the index ulcer were PCR_confirmed as having recurrent ulcers. But the discussion says “A low ― albeit substantial (7%) ― proportion of patients with yaws had serologically confirmed recurrent yaws.” Does that mean the original numbers – 103/1486 – were correct and the 8/12 in active case finding was an anomaly? I am not seeing that you commented on this in the discussion. Is there an advantage to either approach?

7. Discussion: you say that treatment failure is highly unlikely, but could you support that with a reference or data from one of the studies that looked at that? I know that I could look it up but your argument will be stronger with just a sentence with data to support it.

8. Are you surprised at the finding: ‘Nearly half of individuals with recurrent yaws (43%) were children aged 15 years or younger.”? I admit that I am having trouble deriving this based on the Tables, which suggest the majoring of recurrent infections were in <15 year olds.

Reviewer #3: The study presents data on recurrence yaws in a well-followed population, through the use of two distintictive data collection approaches. It is useful data to share with public health institutions working with yaws in PNG and other setttings.

The study is well written and the efforts to bring further understanding on the epidemiology of yaws in Lihir island with existing data are meritable. However, it is no very clear whether looking at retrospective medical registries is the most appropriate method to properly assess recurrence. It is my understanding that the registry was not set up with the objective to monitor cases or recurrence, so the registry should be further explained (i.e. for example in terms of attributes of the surveillance system - see CDC, ECDC surveillance system evaluations). The choice of these two methods, how they complement each other, and their weaknesses for this objective (and the point discussed on fully protective immunity) can be further explained. Thus, it can be better assessed about the methodological implications of measuring recurrence through it.

In terms of recommendations, further reflection on which kind of routine surveillance system would be needed for yaws eradication in the country and in the island would be welcome. Considering the previous work on yaws in the island, authors could propose and advocate on how active follow up could be integrated in this setting

Reviewer #3:

Details on ethical procedures are missing, mainly related to the consent process. The information provided in the study says that “all participants provided informed consent” but it should specify :

- The different consent process for the outpatient clinic participants and for the active surveillance participants. Was informed consent avaliable for conducting a research from the patients included in the passive surveillance data? In case not, it should be specified and whether this has been specifically approved by the local ethical review or exempted from full review.

- To specify how the consent to conduct research was processed for participants below 18 (in both surveillances) – was the caregiver providing with such a consent? Or was the legal responsible of the children. All this should be justified according to PNG law

- Whether consent was oral or written, and in which language was provided.

- Was assent obtained from children to participate (if so, from each age - according to PNG law)?

If those details are included in other publications from this data, please refer to them

PLOS authors have the option to publish the peer review history of their article (what does this mean?). If published, this will include your full peer review and any attached files.

Reviewer #1: Yes: Rafiq N. A. Okine

Reviewer #2: No

Reviewer #3: No
---

## [Decision Letter · Decision Letter 1]

24 Jan 2022

Dear Dr. Gonzalez-Beiras,

We are pleased to inform you that your manuscript 'Yaws recurrence in children at continued risk of infection' has been provisionally accepted for publication in PLOS Neglected Tropical Diseases.

Best regards,

Stuart Robert Ainsworth

Associate Editor

Ana LTO Nascimento

Deputy Editor

Reviewer's Responses to Questions

**Key Review Criteria Required for Acceptance?**

**Methods**

-Are the objectives of the study clearly articulated with a clear testable hypothesis stated?

-Is the study design appropriate to address the stated objectives?

-Is the population clearly described and appropriate for the hypothesis being tested?

-Is the sample size sufficient to ensure adequate power to address the hypothesis being tested?

-Were correct statistical analysis used to support conclusions?

-Are there concerns about ethical or regulatory requirements being met?

Reviewer #1: Subject selection and classification: The authors have provided sufficient information with respect to the queries raised on subject classification particularly in relation to the definitions for recurrence.

Statistical analyses: Good justification was provided for the choice of PWP instead of frailty/random effects model and how within-subject heterogeneity was accounted for.

Ethical considerations: the authors have included some statements on ethical considerations. However with respect to the active surveillance, the statement on all children providing oral consent should be re-considered since ALL children may not have been able to provide oral consent depending on their age. If oral consent was done for specific age group for the active surveillance, this must be stated. Additionally, where assent was provided, this should be clearly stated.

Reviewer #3: (No Response)

**Results**

-Does the analysis presented match the analysis plan?

-Are the results clearly and completely presented?

-Are the figures (Tables, Images) of sufficient quality for clarity?

Reviewer #1: The corrections to the flow chart have been effected and the annotations properly aligned.

The suggestions with respect to the thresholds for the RPR titre levels have been effected 1.8 instead of 1.08.

ALL other corrections effected

Reviewer #3: (No Response)

**Conclusions**

-Are the conclusions supported by the data presented?

-Are the limitations of analysis clearly described?

-Do the authors discuss how these data can be helpful to advance our understanding of the topic under study?

-Is public health relevance addressed?

Reviewer #1: No major revisions to discussions, conclusions or recommendations. Authors have adequately addressed initial review comments.

Reviewer #3: (No Response)

**Editorial and Data Presentation Modifications?**

Reviewer #1: N/A

Reviewer #3: (No Response)

**Summary and General Comments**

Reviewer #1: N/A

Comments provided in initial review

Reviewer #3: (No Response)

PLOS authors have the option to publish the peer review history of their article (what does this mean?). If published, this will include your full peer review and any attached files.

Reviewer #1: **Yes: **Rafiq N.A Okine

Reviewer #3: No

---

## [Editor Report · Acceptance letter]

16 Mar 2022

Dear Dr González-Beiras,

We are delighted to inform you that your manuscript, "Yaws recurrence in children at continued risk of infection," has been formally accepted for publication in PLOS Neglected Tropical Diseases.

Best regards,

Shaden Kamhawi

co-Editor-in-Chief

Paul Brindley

co-Editor-in-Chief
